# Effect of acute alcohol intoxication on mortality, coagulation, and fibrinolysis in trauma patients

Il-Jae Wang[1,2], Byung-Kwan Bae[1], Young Mo Cho[1], Suck Ju Cho[1,3], Seok-Ran Yeom[1,3], Sang-Bong Lee[4], Mose Chun[5], Hyerim Kim[2,6], Hyung-Hoi Kim[2,6], Sun Min Lee[7], Up Huh[8], Soo Young Moon[2,6]*

1 Department of Emergency Medicine, Pusan National University Hospital, Busan, Republic of Korea, 2 Biomedical Research Institute, Pusan National University Hospital, Busan, Republic of Korea, 3 Department of Emergency Medicine, Pusan National University School of Medicine, Gyeongsangnam-do, Yangsan, Republic of Korea, 4 Department of Trauma Surgery, Pusan National University Hospital, Busan, Republic of Korea, 5 Department of Emergency Medicine, Pusan National University Yangsan Hospital, Gyeongsangnam-do, Yangsan, Republic of Korea, 6 Department of Laboratory Medicine, Pusan National University Hospital, Busan, Republic of Korea, 7 Department of Laboratory Medicine, Pusan National University Yangsan Hospital, Gyeongsangnam-do, Yangsan, Republic of Korea, 8 Department of Thoracic and Cardiovascular Surgery, Pusan National University Hospital, Busan, Republic of Korea

* symoon9@gmail.com

**Data Availability Statement:** All relevant data are within the manuscript.

## Abstract

### Background

The effect of alcohol on the outcome and fibrinolysis phenotype in trauma patients remains unclear. Hence, we performed this study to determine whether alcohol is a risk factor for mortality and fibrinolysis shutdown in trauma patients.

### Materials and methods

A total of 686 patients who presented to our trauma center and underwent rotational thromboelastometry were included in the study. The primary outcome was in-hospital mortality. Logistic regression analysis was performed to determine whether alcohol was an independent risk factor for in-hospital mortality and fibrinolysis shutdown.

### Results

The rate of in-hospital mortality was 13.8% and blood alcohol was detected in 27.7% of the patients among our study population. The patients in the alcohol-positive group had higher mortality rate, higher clotting time, and lower maximum lysis, more fibrinolysis shutdown, and hyperfibrinolysis than those in the alcohol-negative group. In logistic regression analysis, blood alcohol was independently associated with in-hospital mortality (odds ratio [OR] 2.578; 95% confidence interval [CI], 1.550–4.288) and fibrinolysis shutdown (OR 1.883 [95% CI, 1.286–2.758]). Within the fibrinolysis shutdown group, blood alcohol was an independent predictor of mortality (OR 2.168 [95% CI, 1.030–4.562]).

**Funding:** This work was supported by a 2-year Research Grant from Pusan National University. No specific other funder is to be disclosed.

**Competing interests:** This work was supported by a 2-year Research Grant from Pusan National University. The authors have declared that no competing interests exist.

## Conclusions

Alcohol is an independent risk factor for mortality and fibrinolysis shutdown in trauma patients. Further, alcohol is an independent risk factor for mortality among patients who experienced fibrinolysis shutdown.

## Introduction

Alcohol is a major risk factor for trauma, and 30–50% of trauma cases involve acute alcohol intoxication [1–3]. Acute alcohol intoxication causes abnormal physiological responses such as impaired cardiovascular response, increased pulmonary vascular resistance, blunted catecholamine release, and coagulation dysfunction [3–8]. Given the fatal effect of coagulopathy in severe cases of trauma, the clinical outcome of alcohol-intoxicated trauma cases is expected to be poor [9]. However, effects of alcohol on the outcome in trauma patients remain unclear, and several studies have shown that the outcome of alcohol intoxicated patients is superior to that of patients who are not intoxicated [2, 10–12].

To further evaluate the relationship between alcohol and coagulopathy, recent studies have used viscoelastic hemostatic assays (VHAs) such as rotational thromboelastometry (ROTEM) and thromboelastography (TEG) [2, 3, 11, 13]. Unlike standard coagulation tests such as prothrombin time (PT) and partial thromboplastin time, which reflect only a small part of the coagulation cascade, VHAs have the advantage of demonstrating the entire process of coagulation, from clot generation to fibrinolysis, in real time [14]. Recent studies using VHAs have shown that alcohol impairs clot formation and inhibits fibrinolysis [2]. Moreover, an association between high blood alcohol and fibrinolysis shutdown was also observed [15].

Fibrinolysis shutdown is a state of impaired fibrinolysis. Fibrinolysis shutdown is the most common fibrinolysis phenotype in severe trauma cases and is associated with higher mortality than that in physiologic fibrinolysis [16]. However, only few studies have been conducted on whether alcohol intoxication is associated with fibrinolysis shutdown, and no studies have been conducted on the clinical outcome of alcohol-intoxicated trauma patients with fibrinolysis shutdown. As fibrinolysis shutdown and alcohol intoxication are both common in trauma patients, the mutual correlation between these phenomena and prognosis are not clear.

The objectives of this study were to determine (a) the effect of alcohol on mortality and coagulation in trauma patients, (b) whether alcohol is a risk factor of fibrinolysis shutdown, and (c) the influence of alcohol on mortality in case of fibrinolysis shutdown.

## Materials and methods

### Study design and setting

This is a retrospective single-center study involving an analysis of data collected from a registry. Our trauma center serves as a Level 1 regional trauma center and is one of the largest trauma centers in our country. Approximately 1,000 patients with an injury severity score (ISS) of 15 or higher present to the trauma center each year. The study protocol was approved by the Institutional Review Board of Pusan National University Hospital (IRB-2007-033-093). The patients' information was anonymously analyzed; therefore, the requirement for informed consent was waived.

### Study population

Patients who presented to the trauma center and underwent the ROTEM test between January 2016 and December 2019 were included. The exclusion criteria were (1) age ≤15 years, (2) cardiac arrest, and (3) no blood alcohol test performed.

## Data collection and variables

Data were extracted from the Korea Trauma Database (KTDB) and electronic medical records of our hospital. The KTDB was established in 2013 by the Ministry of Health and Welfare of the Republic of Korea for collecting detailed information on trauma patients nationwide [17]. Data of the following variables were collected from the database: age, sex, vital signs, and Glasgow Coma Scale (GCS) score at the time of presentation to the trauma center, mechanism of injury, ISS, head Abbreviated Injury Scale (AIS), thorax AIS, abdomen AIS, substantial bleeding, massive transfusion (MT), and in-hospital mortality. We also collected the following laboratory data: prothrombin time international normalized ratio (PT INR), lactic acid level, platelet count, and blood pH. Blood alcohol (ethanol) was tested in the majority of severely traumatic patients by using Roche diagnostics instruments and reagents (Rotkreuz, Switzerland). Blood alcohol tests were not conducted if too much time had passed since the occurrence of the injury based on the attending physician's judgement. VHA was performed using ROTEM delta (TEM International GmbH, Munich, Germany). The necessity for performing ROTEM test was determined by the attending physician in charge based on hemodynamic signs, FAST results, and injury mechanism at our trauma center. The collected ROTEM data included extrinsically-activated TEM (EXTEM) clotting time (CT), EXTEM clot formation time (CFT), EXTEM maximum clot firmness (MCF), and EXTEM maximum lysis (ML). Hyperfibrinolysis was defined as the presence of EXTEM ML $\geq$15% and fibrinolysis shutdown as EXTEM ML $\leq$3% [16, 18]. For ROTEM and laboratory analysis, blood samples were drawn within 15 min of initial presentation to the trauma center and ROTEM samples were collected before tranexamic acid (TXA) was administered.

The primary outcome was in-hospital mortality. Time to death was defined as a time period (hr) from the injury time to the in hospital death. The secondary outcomes were substantial bleeding and MT, which were respectively defined as $\geq$5 units packed red blood cells transfused within 4 h of presentation and $\geq$10 units within 24 h at the trauma center, respectively [19].

## Statistical analysis

Categorical variables were reported as frequencies and percentages, whereas continuous variables were reported as medians and interquartile ranges (IQR). None of the continuous variables showed Gaussian distribution in the Shapiro–Wilk test. We compared categorical variables using Fisher's exact test and continuous variables using Mann–Whitney U test. Logistic regression analysis was performed to evaluate the independent effect of alcohol on mortality and fibrinolysis shutdown. MedCalc version 19.4 software (Ostend, Belgium) was used for statistical analysis. P values were two-sided, and P<0.05 was considered statistically significant.

# Results

## Patients' characteristics

During the study period, 837 patients with ROTEM data were included. Based on the exclusion criteria, 151 patients were excluded due to lack of blood alcohol test (n = 111), age $\leq$15 years (n = 11), and presentation to the hospital with cardiac arrest (n = 29). Finally, 686 patients were included in this study (Fig 1).

This included 525 men (76.5%). The median age was 55 years (IQR 39–66 years) and median ISS was 22 (16–29). In-hospital mortality was 95 (13.8%), and 133 (19.4%) patients

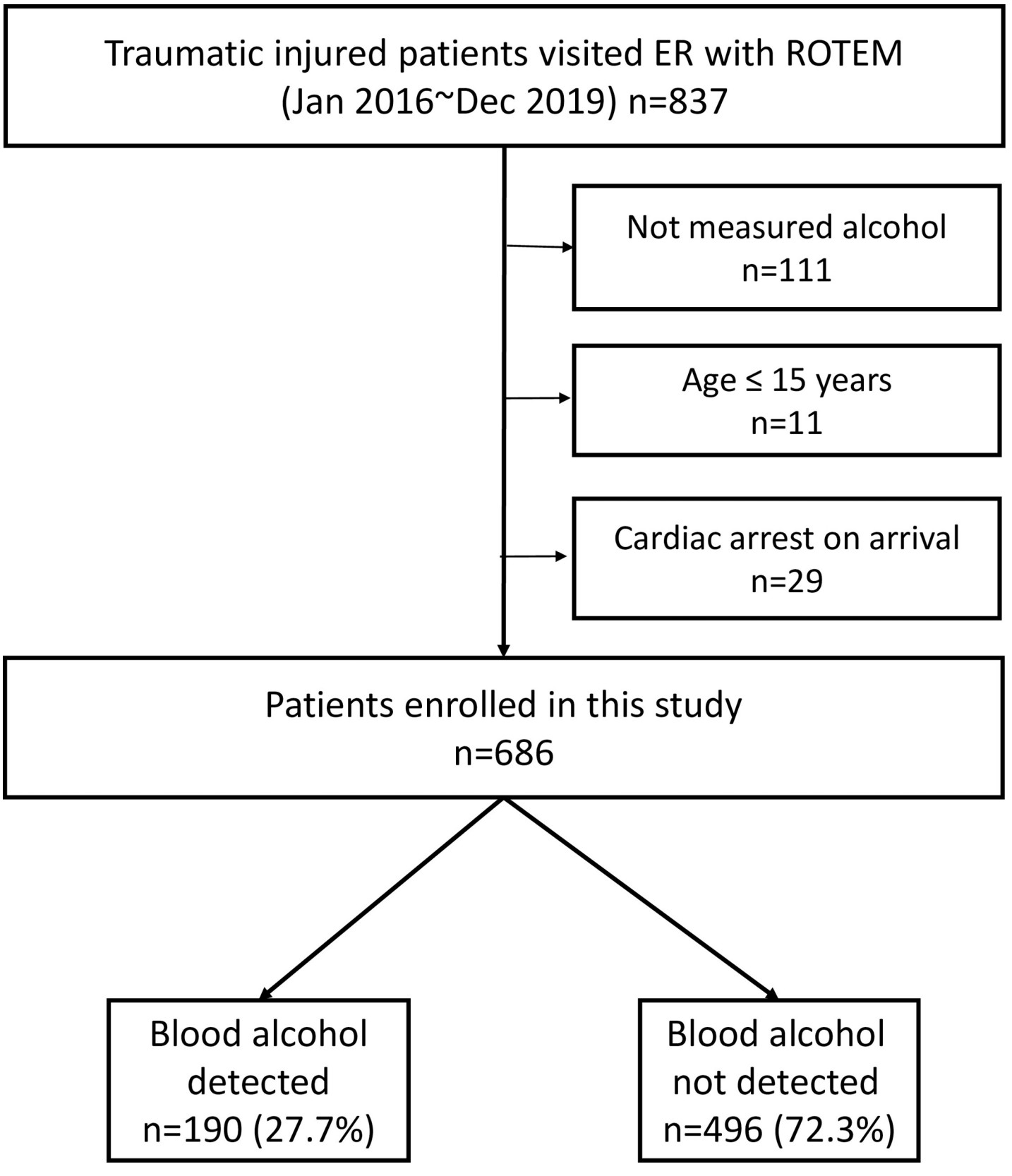

**Fig 1. Flow chart of enrolled patients in this study.**

**Table 1. Characteristics of enrolled patients according to the detection of blood alcohol.**

| | Total (n = 686) | Blood alcohol not detected (n = 496) | Blood alcohol detected (n = 190) | *P* value |
|---|---|---|---|---|
| Age, year | 55 (39–66) | 57 (44.5–69) | 47 (32–58) | <0.001 |
| Male, n | 525 (76.5%) | 364 (73.4%) | 161 (84.7%) | 0.002 |
| Mechanism of injury, n | | | | |
| Fall from height | 157 (22.9%) | 112 (22.6%) | 45 (23.7%) | 0.759 |
| Pedestrian collision | 150 (21.9%) | 100 (20.2%) | 50 (26.3%) | 0.084 |
| Driver and passenger collision | 146 (21.3%) | 120 (24.2%) | 26 (13.7%) | 0.003 |
| Motorcycle collision | 93 (13.6%) | 61 (12.3%) | 32 (16.8%) | 0.124 |
| Penetrating | 37 (5.4%) | 19 (3.8%) | 18 (9.5%) | 0.764 |
| Ground level fall | 24 (3.5%) | 17 (3.4%) | 7 (3.7%) | 0.848 |
| Bicycle collision | 20 (2.9%) | 17 (3.4%) | 3 (1.6%) | 0.209 |
| SBP, mmHg | 100 (70–120) | 100 (80–130) | 90 (70–110) | <0.001 |
| GCS | 15 (8–15) | 15 (11–15) | 11.5 (5–15) | <0.001 |
| ISS | 22 (16–29) | 22 (14–29) | 25 (17–34) | <0.001 |
| Head & neck AIS $\geq$ 3, n | 202 (29.4%) | 124 (25.0%) | 78 (41.1%) | <0.001 |
| Thorax AIS $\geq$ 3, n | 302 (44.0%) | 214 (43.1%) | 88 (46.3%) | 0.454 |
| Abdomen AIS $\geq$ 3, n | 192 (28.0%) | 134 (27.0%) | 58 (30.5%) | 0.36 |
| PT INR | 1.1 (1.0–1.2) | 1.1 (1.0–1.2) | 1.1 (1.0–1.3) | 0.275 |
| Lactic acid, mmol/L | 3.3 (2.0–5.7) | 2.7 (1.7–4.7) | 5.3 (3.4–7.7) | <0.001 |
| Platelet, ×1,000/μL | 214 (170–259) | 209.5 (168–254) | 228 (181–279) | 0.001 |
| pH | 7.4 (7.3–7.4) | 7.4 (7.3–7.4) | 7.3 (7.2–7.4) | <0.001 |
| EXT_CT, sec | 58 (52–71) | 57 (51–68) | 61 (54–75) | 0.005 |
| EXT_CFT, sec | 93 (74–122) | 93 (72–124) | 96 (79–121) | 0.124 |
| EXT_MCF, mm | 60 (54–64) | 60 (54.5–64) | 60 (53–63) | 0.13 |
| EXT_ML, % | 5 (2–8) | 5 (3–8) | 4 (1–8) | 0.028 |
| EXT_ML phenotype, n | | | | 0.001 |
| • Physiologic (Others) | 352 (51.3%) | 276 (55.6%) | 76 (40.0%) | |
| • Shutdown (ML $\leq$ 3%) | 274 (39.9%) | 183 (36.9%) | 91 (47.9%) | |
| • Hyperfibrinolysis (ML $\geq$ 15%) | 60 (8.7%) | 37 (7.5%) | 23 (12.1%) | |
| In hospital death, n | 95 (13.8%) | 53 (10.7%) | 42 (22.1%) | <0.001 |
| Time from injury to death, hr | 39 (12–122) | 54 (16–199) | 34 (8–71) | 0.056 |
| Substantial bleeding, n ($\geq$ 5 RBCs within 4 hr) | 174 (25.4%) | 111 (22.4%) | 63 (33.2%) | 0.004 |
| Massive transfusion, n ($\geq$ 10 RBCs within 24 hr) | 133 (19.4%) | 88 (17.7%) | 45 (23.7%) | 0.098 |

Data was presented as number (percentage) or median (interquartile range). Statistical testing was performed with Mann Whitney U test or Fisher's exact test.
Abbreviations: SBP, systolic blood pressure; GCS, Glasgow Coma Scale; ISS, injury severity score; AIS, abbreviated injury score; PT INR, prothrombin time international normalized ratio; EXT, ROTEM EXTEM.

received MT. Blood alcohol was detected (>10.1 mg/dL) in 190 patients (27.7%), and the median blood alcohol concentration in these patients was 182 mg/dL (122–243.9).

## Comparison of the alcohol-positive and alcohol-negative groups

Patients were divided into alcohol-positive and alcohol-negative groups based on the detection limit of 10.1 mg/dL (Table 1) and their characteristics were then compared. This cutoff was determined by the manufacturer, Roche diagnostics, as values of ETOH below 10.1 mg/dL were not recorded as a numerical data. Hence, this cutoff value intended by the manufacturer was adopted in our study to categorize the patients. Patients in the alcohol-positive group were significantly younger (P<0.001) and had male predominance (P = 0.002) as compared to those

in the alcohol-negative group. Patients in the alcohol-positive group also had a significantly lower systolic blood pressure (SBP) (100 vs. 90, P<0.001), lower GCS score (15 vs. 11.5, P<0.001), higher ISS (22 vs. 25, P<0.001), and higher lactic acid levels (2.7 vs. 5.3, P<0.001) than those in the alcohol-negative group. Further, patients in the blood alcohol positive group had higher head and neck AIS, but not thorax and abdomen AIS than those in the blood alcohol negative group. In-hospital mortality rate was significantly higher in the alcohol-positive group than that in the alcohol-negative group (10.7% vs. 22.1%, P<0.001). In addition, the number of patients with substantial bleeding was higher in the alcohol-positive group than that in the alcohol negative group (22.4% vs. 33.2%, P = 0.004). Regarding ROTEM, no significant difference was observed in EXTEM CFT and MCF. Patients in the alcohol-positive group had higher EXTEM CT (P = 0.005) and lower EXTEM ML (P = 0.028) than those in the alcohol-negative group. The fibrinolysis phenotypes were significantly different between the two groups (P = 0.001). The alcohol-positive group had more fibrinolysis shutdown and hyperfibrinolysis than the alcohol-negative group. To determine whether alcohol was an independent risk factor for in-hospital mortality, logistic regression analysis was performed (Table 2). Blood alcohol detection remained independently associated with in-hospital mortality after adjustment for age, sex, and ISS (odds ratio [OR] 2.578; 95% confidence interval [CI], 1.550–4.288; P<0.001).

## Comparison of patients according to fibrinolysis phenotype

Most patients had physiologic fibrinolysis (51.3%), some had fibrinolysis shutdown (39.9%) and hyperfibrinolysis (8.7%). Differences in age and sex were not significant among the groups. The in-hospital mortality rate was significantly different between the groups (P<0.001). The hyperfibrinolysis group showed the highest mortality rate (48.3%) and the fibrinolysis shutdown group (14.6%) showed approximately two times higher mortality than that of the physiologic fibrinolysis group (7.4%). In the aspect of bleeding and transfusion, the hyperfibrinolysis group showed the highest frequency of MT and substantial bleeding, and the fibrinolysis shutdown group showed approximately two times higher MT rate than that of physiologic fibrinolysis group. In the fibrinolysis shutdown group, alcohol intoxication (33.2%) was more frequently observed than that in the physiologic fibrinolysis group (21.6%). Table 3 summarizes the characteristics of patients according to fibrinolysis phenotype. To identify whether alcohol is an independent risk factor for fibrinolysis shutdown compared with physiologic fibrinolysis, a logistic regression analysis was performed by adjusting for age and ISS. Logistic regression analysis revealed that alcohol was independently associated with fibrinolysis shutdown (OR 1.883 [95% CI, 1.286–2.758], P<0.001).

## Effect of alcohol on mortality in fibrinolysis shutdown patients

In the subgroup analysis, we compared the alcohol-positive and alcohol-negative patients in the fibrinolysis shutdown group (Table 4). Similar to the study population, alcohol-positive patients were younger and died more frequently in the fibrinolysis shutdown group. ISS was

**Table 2. Logistic regression to predict in hospital death.**

|  | Odds ratio | 95% CI | P value |
|---|---|---|---|
| Age | 1.0310 | 1.0162 to 1.0459 | <0.001 |
| Male | 1.8560 | 0.9946 to 3.4637 | 0.052 |
| ISS | 1.0708 | 1.0485 to 1.0936 | <0.001 |
| Blood alcohol ≥10.2 mg/dL | 2.5787 | 1.5508 to 4.2880 | <0.001 |

**Table 3. Characteristics of enrolled patients according to the fibrinolysis status.**

| | Physiologic fibrinolysis (n = 352) | Shutdown (n = 274) | Hyperfibrinolysis (n = 60) | P value |
|---|---|---|---|---|
| Age, year | 53 (38–64) | 57 (41–69) | 54 (43–66) | 0.099 |
| Male, n | 272 (77.3%) | 206 (75.2%) | 47 (78.3%) | 0.781 |
| ISS | 22.0 (14.0–27.0) | 25.0 (17.0–33.0) | 29.0 (19.5–41.5) | <0.001 |
| Blood alcohol ≥10.2 mg/dL | 76 (21.6%) | 91 (33.2%) | 23 (38.3%) | <0.001 |
| Lactic acid, mmol/L | 2.8 (1.7–4.4) | 3.9 (2.3–6.2) | 7.6 (4.5–11.5) | <0.001 |
| In house death, n | 26 (7.4%) | 40 (14.6%) | 29 (48.3%) | <0.001 |
| Substantial bleeding, n (≥ 5 RBCs within 4 hr) | 59 (16.8%) | 77 (28.1%) | 38 (63.3%) | <0.001 |
| Massive transfusion, n (≥ 10 RBCs within 24 hr) | 40 (11.4%) | 57 (20.8%) | 36 (60.0%) | <0.001 |

Data in parenthesis was presented as percentage or interquartile range. Statistical testing was performed with Mann Whitney U test or Fisher's exact test.

higher in the alcohol-positive group compared to the alcohol-negative group. However, it was not significant. To determine whether alcohol was still an independent risk factor in patients with fibrinolysis shutdown, we performed logistic regression analysis to predict mortality. After adjustment for age, sex, and ISS, blood alcohol was identified as an independent predictor of mortality in the fibrinolysis shutdown group (OR 2.168 [95% CI, 1.030–4.562], P = 0.0416) (Table 5).

## Discussion

The main results of this study were (a) alcohol intoxication was associated with prolonged CT and decreased fibrinolysis, (b) in-hospital mortality in the alcohol-positive group was significantly higher than that of the alcohol-negative group, and alcohol was identified as an independent risk factor of mortality, (c) fibrinolysis shutdown was independently associated with blood alcohol, (d) the alcohol-positive group was associated with significantly higher in-hospital mortality rate than the alcohol-negative group in the fibrinolysis shutdown group.

Alcohol is considered as a major risk factor for trauma. It impairs multiple compensatory responses to trauma, which are crucial for the patient outcome [4, 7]. The most critical effect of alcohol in trauma patients is impaired coagulation [3]. Hemorrhage is the leading cause of preventable deaths in trauma patients [9], and with acute traumatic coagulopathy the mortality rate of bleeding patients significantly increases [20]. Previous studies have shown that coagulation impairment due to alcohol cannot be identified with the standard coagulation test, but it can be identified using VHA [2, 3, 21]. Similar to the previous studies, we reported that prothrombin time INR was not different in the alcohol-positive and alcohol-negative groups, but EXTEM CT was significantly different.

**Table 4. Effect of blood alcohol in fibrinolysis shutdown patients.**

| | Blood alcohol not detected (n = 183) | Blood alcohol detected (n = 91) | P value |
|---|---|---|---|
| Age, year | 60 (47.5–73) | 48 (34–61.5) | <0.001 |
| Male, n | 127 (69.4%) | 79 (86.8%) | 0.003 |
| SBP, mmHg | 100 (80–130) | 90 (70–110) | 0.019 |
| ISS | 24 (17–33) | 26 (19–34) | 0.085 |
| PT INR | 1.1 (1.0–1.3) | 1.1 (1.0–1.3) | 0.349 |
| Lactic acid, mmol/L | 3.0 (1.9–5.3) | 5.4 (4.0–7.8) | <0.001 |
| EXT_CT, sec | 62 (53–77.5) | 64 (55–79.5) | 0.194 |
| In house death, n | 20 (10.9%) | 20 (22.0%) | 0.024 |

**Table 5. Logistic regression to predict in-hospital death among shutdown patients.**

|  | Odds ratio | 95% CI | P value |
|---|---|---|---|
| Age | 1.0146 | 0.9939 to 1.0358 | 0.1688 |
| Male | 2.659 | 0.9361 to 7.5523 | 0.0663 |
| ISS | 1.056 | 1.0236 to 1.0895 | 0.0006 |
| Blood alcohol ≥10.2 mg/dL | 2.1677 | 1.0300 to 4.5621 | 0.0416 |

The unusual feature of coagulation impairment due to alcohol is that it is not correlated with clinical results [2, 3, 7, 10, 12]. Howard et al. conducted a single-center study using TEG on 264 severely traumatized patients at a Level 1 trauma center [3]. They found that although the time for initial clot formation was prolonged in the blood alcohol-positive group, there was no significant difference in mortality and multiorgan failure. In subsequent studies conducted by the same group, impairment of clot generation and inhibition of fibrinolysis caused by alcohol were found [2]. This bidirectional effect was suggested as the reason for the lack of correlation between coagulation impairment and clinical outcome.

The result of our study is consistent with previous findings. Our results demonstrated that alcohol-positive patients showed significantly prolonged CT and decreased ML. Unlike previous studies, our study shows that the mortality rate of the alcohol-positive group was significantly higher than that of the alcohol-negative group. Furthermore, alcohol was identified as an independent risk factor for mortality after adjustment for age, sex, and ISS. This meant that alcohol-intoxicated patients were mostly men, young, and severely injured, but the alcohol itself could contribute to an unfavorable prognosis. The reason is not clear, but one possible explanation is the heterogeneous pathological fibrinolysis phenotypes, such as hyperfibrinolysis and fibrinolysis shutdown, in the alcohol-positive group.

Fibrinolysis is an important component of acute traumatic coagulopathy and pathologic fibrinolysis condition worsens patient outcome [16, 18, 22]. Hyperfibrinolysis is a state wherein fibrinolysis is pathologically activated and overwhelms fibrin formation. Hyperfibrinolysis is the least frequently observed phenotype among the three fibrinolysis phenotypes, and has the highest mortality [23–25]. Fibrinolysis shutdown is a contrasting phenomenon of hyperfibrinolysis, in which fibrinolysis is abnormally reduced. Similar to previous findings, our study showed that hyperfibrinolysis had an overwhelmingly higher mortality rate than the other phenotypes. In contrast, fibrinolysis shutdown had a lower mortality rate than hyperfibrinolysis. However, it was higher than the mortality rate of physiologic fibrinolysis.

Recently, the importance of accurate diagnosis of fibrinolysis shutdown has emerged. This is because fibrinolysis shutdown has a high incidence rate and may worsen the patient's condition when TXA is administered in a patient with fibrinolysis shutdown [16, 23, 24]. In 2020, Stettler et al. demonstrated blood alcohol to be associated with lower fibrinolysis and identified high blood alcohol as an independent predictor of fibrinolysis shutdown [15]. In our study, blood alcohol was identified as an independent risk factor for fibrinolysis shutdown. Furthermore, the effects of alcohol on mortality among patients with fibrinolysis shutdown were analyzed, and alcohol was an independent risk factor of mortality in such patients. To the best of our knowledge, this is the first study to identify the relationship between alcohol and outcomes in patients with fibrinolysis shutdown. Although the cause is not clear, one hypothesis implies that alcohol worsens fibrinolysis shutdown because it inhibits fibrinolysis. There are several reports on the association of acute alcohol intake and marked inhibition of fibrinolysis, both *in vitro* and in healthy subjects [8, 26]. The exact mechanism of such a phenomenon must be elucidated through further research.

In our study, patients in the alcohol positive group had lower GCS, lower SBP, higher head AIS and higher lactic acid levels compared to those in the alcohol negative group. We believe that the reason for lower GCS was not just alcohol, but also the effects of head trauma and the state of shock. Alcohol has the potential to induce metabolic acidosis. During the metabolism of alcohol, pyruvate is reduced to lactate, which creates lactic acidosis. Hence, it is challenging to distinguish the cause of lactate elevation in a trauma patient who consumed alcohol, as it could be due to high degree of shock or metabolism of the alcohol. However, further studies are required to devise methods to identify the cause of lactate elevation in such patients.

Our study has several limitations. First, this was a retrospective study; a potential bias could have existed. Second, it was a single-center study; hence, the generalizability of our results may be uncertain because of the variation of regional features. Third, alcohol can be evaluated as a continuous variable, but we analyzed it as a categorical variable. Fourth, the necessity for performing ROTEM test was decided by the physician based on hemodynamic sign, ultrasound results, and injury mechanism. Hence, the possibility of a selection bias cannot be totally excluded.

## Conclusions

In conclusion, we identified that alcohol impairs initial clot generation, inhibits fibrinolysis, and is an independent risk factor for fibrinolysis shutdown in trauma patients. Further, in patients with fibrinolysis shutdown, alcohol is an independent risk factor for mortality.

## Acknowledgments

The authors would like to thank all the patients and their families who were included in this study.

## Author Contributions

**Conceptualization:** Il-Jae Wang.

**Data curation:** Il-Jae Wang, Soo Young Moon.

**Formal analysis:** Soo Young Moon.

**Funding acquisition:** Seok-Ran Yeom.

**Investigation:** Il-Jae Wang, Byung-Kwan Bae, Young Mo Cho, Suck Ju Cho, Seok-Ran Yeom, Sang-Bong Lee, Mose Chun, Up Huh.

**Methodology:** Hyerim Kim, Hyung-Hoi Kim, Sun Min Lee.

**Resources:** Byung-Kwan Bae, Young Mo Cho, Suck Ju Cho, Seok-Ran Yeom, Sang-Bong Lee, Mose Chun, Up Huh.

**Software:** Soo Young Moon.

**Visualization:** Soo Young Moon.

**Writing – original draft:** Il-Jae Wang.

**Writing – review & editing:** Soo Young Moon.

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
