## [Decision Letter · Decision Letter 0]

22 Jan 2021

PONE-D-20-38577

Effect of Acute Alcohol Intoxication on Mortality, Coagulation, and Fibrinolysis in Trauma Patients

PLOS ONE

Dear Dr. Moon,

Thank you for submitting your manuscript to PLOS ONE. After careful consideration, we feel that it has merit but does not fully meet PLOS ONE’s publication criteria as it currently stands. Therefore, we invite you to submit a revised version of the manuscript that addresses the points raised during the review process.

We look forward to receiving your revised manuscript.

Kind regards,

Zsolt J. Balogh, MD, PhD, FRACS

Academic Editor

PLOS ONE

Journal Requirements:

"This work was supported by a 2-year Research Grant from Pusan National University."

Reviewers' comments:

Reviewer's Responses to Questions

**Comments to the Author**

1. Is the manuscript technically sound, and do the data support the conclusions?

Reviewer #1: Partly

Reviewer #2: Yes

2. Has the statistical analysis been performed appropriately and rigorously? 

Reviewer #1: Yes

Reviewer #2: Yes

3. Have the authors made all data underlying the findings in their manuscript fully available?

Reviewer #1: No

Reviewer #2: Yes

4. Is the manuscript presented in an intelligible fashion and written in standard English?

Reviewer #1: Yes

Reviewer #2: Yes

5. Review Comments to the Author

Reviewer #1: The authors have performed a retrospective analysis of prospectively collected data evaluating the association of ETOH intoxication with mortality and ROTEM parameters in trauma patients. The authors show that ETOH intoxication is associated with increased clotting time, decreased lysis and increased mortality in their cohort. I have the following comments and questions for the authors:

1. What were the indications to perform ROTEMs in trauma patients during the course of the study? Essentially, what were the inclusion criteria for the study? This is a busy trauma center and over a 3 year period only 686 patients were included indicating a certain population was targeted.

2. Was there a difference in mechanism of injury between the alcohol positive group and alcohol negative group? Mechanism of injury should be included in the statistical analysis because it could affect outcome. Similarly, was there a difference in AIS regions injured? The alcohol positive patients were more likely to have a low GCS, did they present more commonly with TBI which could explain all of the results of the study? To best answer this question, the authors should provide cause of death comparison between groups. The authors should discuss whether the lower GCS in the ETOH group was due to intoxication versus brain injury versus shock which could all cause a decreased GCS.

3. Patients who were positive for alcohol presented with a higher lactate level. This could indicate that they presented with a higher degree of shock or it could be secondary to alcohol consumption. This requires a discussion from the authors and an opinion.

4. There is no clear association between administration of TXA and increased mortality in patients with fibrinolysis shutdown. In fact, randomized trials of TXA in trauma patients do not suggest it causes fibrinolysis shutdown. The comment about TXA should be removed from the conclusions because the authors have no data to substantiate the statement.

Reviewer #2: The authors have compared ETOH, ROTEM and outcomes is a selected group of trauma patients, and correlated lab values with ETOH. I have several comments

1. the authors admit > 1000 severely injured patients a year, but over 4 years only have rotem data on 837, what was the criteria to run the rotem test?

2, why was the cut of value of (>10.1 mg/dL) used for etoh ?

3. what was the criteria used for measuring ETOH?

4. when were the samples drawn?

5. the authors definition of MT, > 10 units of RBCs in 24 hrs, is outdated and perpetuates the serious issue of survival bias. Especially as this is a study of coagulation Please use a more modern definition , eg 3 units in 6 hrs or something similar.

6. the real issue here, is there an association between ETH and more bleeding, rather than just abnormal lab values. The association between etoh and lab values has been shown many times.

7. the authors associate ETOH and mortality, but what was the cause of death in the patients and when did they die? this issue is really important as the authors have an opportunity to tease out this important issue, as they do see an association between etoh use, rotem and mortality.

8. How many patients and in what groups was TXA used, and was it given before the ROTEM sample collected?

6. PLOS authors have the option to publish the peer review history of their article (what does this mean?). If published, this will include your full peer review and any attached files.

Reviewer #1: **Yes: **Martin A. Schreiber, MD FACS FCCM

Reviewer #2: No

---

## [Author Response · Author response to Decision Letter 0]

5 Mar 2021

Please see the attached file named "reply1_editage_final".

---

## [Decision Letter · Decision Letter 1]

8 Mar 2021

Effect of acute alcohol intoxication on mortality, coagulation, and fibrinolysis in trauma patients

PONE-D-20-38577R1

Dear Dr. Moon,

We’re pleased to inform you that your manuscript has been judged scientifically suitable for publication and will be formally accepted for publication once it meets all outstanding technical requirements.

Kind regards,

Zsolt J. Balogh, MD, PhD, FRACS

Academic Editor

PLOS ONE

Additional Editor Comments (optional):

Reviewers' comments:

Reviewer's Responses to Questions

**Comments to the Author**

1. If the authors have adequately addressed your comments raised in a previous round of review and you feel that this manuscript is now acceptable for publication, you may indicate that here to bypass the “Comments to the Author” section, enter your conflict of interest statement in the “Confidential to Editor” section, and submit your "Accept" recommendation.

Reviewer #1: All comments have been addressed

Reviewer #2: All comments have been addressed

2. Is the manuscript technically sound, and do the data support the conclusions?

Reviewer #1: Yes

Reviewer #2: (No Response)

3. Has the statistical analysis been performed appropriately and rigorously? 

Reviewer #1: Yes

Reviewer #2: (No Response)

4. Have the authors made all data underlying the findings in their manuscript fully available?

Reviewer #1: Yes

Reviewer #2: (No Response)

5. Is the manuscript presented in an intelligible fashion and written in standard English?

Reviewer #1: Yes

Reviewer #2: (No Response)

6. Review Comments to the Author

Reviewer #1: The authors have addressed my comments except for cause of death which is information they do not have. The manuscript remains a good addition to the literature.

Reviewer #2: (No Response)

7. PLOS authors have the option to publish the peer review history of their article (what does this mean?). If published, this will include your full peer review and any attached files.

Reviewer #1: **Yes: **Martin A. Schreiber MD FACS FCCM

Reviewer #2: No

---

## [Editor Report · Acceptance letter]

15 Mar 2021

PONE-D-20-38577R1 

Effect of acute alcohol intoxication on mortality, coagulation, and fibrinolysis in trauma patients 

Dear Dr. Moon:

I'm pleased to inform you that your manuscript has been deemed suitable for publication in PLOS ONE. Congratulations! Your manuscript is now with our production department. 

Kind regards, 

on behalf of

Dr. Zsolt J. Balogh 

Academic Editor

PLOS ONE